# The “Balloon-Like” Sign: Differential Diagnosis between Postoperative Air Leak and Residual Pleural Space: Radiological Findings and Clinical Implications of the Young–Laplace Equation

**DOI:** 10.3390/cancers14143533

**Published:** 2022-07-20

**Authors:** Francesco Petrella, Stefania Rizzo, Luca Bertolaccini, Monica Casiraghi, Lara Girelli, Giorgio Lo Iacono, Antonio Mazzella, Lorenzo Spaggiari

**Affiliations:** 1Department of Thoracic Surgery, IEO European Institute of Oncology, IRCCS, 20141 Milan, Italy; luca.bertolaccini@ieo.it (L.B.); monica.casiraghi@ieo.it (M.C.); lara.girelli@ieo.it (L.G.); giorgio.loiacono@ieo.it (G.L.I.); antonio.mazzella@ieo.it (A.M.); lorenzo.spaggiari@ieo.it (L.S.); 2Department of Oncology and Hemato-Oncology, Università degli Studi di Milano, 20141 Milan, Italy; 3Department of Radiology, Ente Ospedaliero Cantonale (EOC) Istituto di Imaging della Svizzera Italiana (IIMSI), 6903 Lugano, Switzerland; stefaniamariarita.rizzo@eoc.ch; 4Facoltà di Scienze Biomediche, Università della Svizzera Italiana, Via Buffi 13, 6900 Lugano, Switzerland

**Keywords:** residual pleural space, postoperative air leaks, Young–Laplace equation

## Abstract

**Simple Summary:**

Postoperative residual pleural space and postoperative air leaks after lung resection are two different clinical entities requiring completely different approaches. Residual postoperative pleural space is a part of the pleural cavity that is not fully reoccupied by the remaining lung after pulmonary resection. No treatment is needed in the asymptomatic residual pleural space without any persistent air leak, and chest drain removal can be safely planned. On the contrary, an active and prolonged air leak after lung resection is an absolute contraindication to chest drain removal that may culminate in hypertensive pneumothorax, subcutaneous emphysema, and severe respiratory symptoms. In order to further contribute to an appropriate differential diagnosis between these two settings, we propose a radiological sign that is observed only in the case of residual plural space. In this case, in fact, the lung takes the form of a round balloon due to the hyperinflation condition, which is governed by the Young–Laplace equation describing the capillary pressure difference sustained across the interface between two static fluids, such as water and air, due to the phenomenon of wall tension.

**Abstract:**

In this paper, we propose a radiological sign for an appropriate differential diagnosis between postoperative pleural space and active air leak after lung resection. In the case of residual pleural space without any active air leak, the lung takes the form of a round balloon due to the hyperinflation condition, which is governed by the Young–Laplace equation describing the capillary pressure difference sustained across the interface between two static fluids, such as water and air, due to the phenomenon of wall tension. The two principal mechanisms by which a lung forms a spherical image are shear-controlled detachment induced by shear stress on the membrane surface, and spontaneous detachment induced by a gradient in Young–Laplace pressure. On the contrary, the lung maintains its tapered shape in the case of an active air leak because the continuous air refill does not allow a complete parenchyma re-expansion.

## 1. Introduction

Residual postoperative pleural space is a part of the pleural cavity that is not fully reoccupied by the remaining lung after pulmonary resection. It is a widespread problem after lung resection, and its clinical implications may range from a completely asymptomatic postoperative course to pleural empyema requiring surgical therapy. It is reported to occur in about 20% of patients receiving anatomical lung resection; in this group, 40% of cases are bilobectomies or lobectomies and wedge resections, while 5%–10% of cases are segmentectomies or wedge resections alone [1].

No treatment is needed in the asymptomatic residual pleural space without any persistent air leak, and chest drain removal can be safely planned [2,3]. On the contrary, an active and prolonged air leak after lung resection is an absolute contraindication to chest drain removal that may culminate in hypertensive pneumothorax, subcutaneous emphysema, and severe respiratory symptoms [4,5,6].

In some cases of residual pleural space without persistent air leak, pleural drain systems may mimic an active air leak—as witnessed by an air bubble in the drain chamber—that is due to the pleural space effect, which can be indistinguishable from an active air leak on checking a traditional water seal chest drain system [3]. Although it has been reported that a digital chest drain system might be helpful in identifying a pleural space effect, preventing the misinterpretation of active air leaks and allowing safe chest drain removal at the right time [3], the safest approach would be to clamp the chest tube for at least 24–48 h and repeat the chest X-ray to monitor the expansion of the lung; only in the case of the absence of lung expansion modification is chest drain removal then recommended.

Here, we report a radiological sign that helps to distinguish between postoperative pleural space and active air leak.

## 2. Physical Considerations

In the case of residual pleural space without any active air leak, the lung takes the form of a round balloon (Figure 1a1,a2,b) due to the hyperinflation condition, which is governed by the Young–Laplace equation describing the capillary pressure difference sustained across the interface between two static fluids, such as water and air, due to the phenomenon of wall tension [7,8,9].

Inflating the alveoli during the respiratory process requires increased internal pressure relative to the surrounding environment. This is accomplished by lowering the pressure in the thoracic cavity below that of the surrounding atmosphere. The required amount of net pressure for inflation is determined by the surface tension and radius of the tiny balloon-like alveoli. One of the standard, well-known forms of the Young–Laplace equation is the following:$$∆p=-\sigma C_{f}=-\sigma \left(\frac{1}{r_{1}}+\frac{1}{r_{2}}\right)$$
where Δ*p* is the Laplace pressure (the external pressure minus the internal pressure), *σ* is the surface tension (or wall tension), *C_f_* is the mean curvature, and *r*_1_ and *r*_2_ are the principal radii of curvature. This law essentially states that the pressure inside an elastic sphere is inversely proportional to the radius, assuming that the surface tension remains constant. According to this relationship, smaller spheres with smaller radii will have a higher transmural pressure at any given surface tension value, i.e., they will want to collapse the larger spheres. The implications of this law for alveoli are that small alveoli (collapsed and nearly collapsed) are more difficult to inflate than large alveoli, which contributes to the low compliance observed at small lung volumes; smaller alveoli will accelerate their demise by emptying into larger adjacent alveoli; and the filtration of fluid across the pulmonary capillary wall is dependent on the hydrostatic pressure gradient, which is increased by alveolar surface tension [10]. Within the lungs, as an adjunctive variable, the alveoli are internally coated by surfactant, whose concentration increases proportionately to the increasing pressures generated by their walls when deflating, which eventually allows the lungs to function normally [11,12].

Additionally, the Young–Laplace equation could describe the pressure by which the lung could form a balloon-like appearance:$$∆p=P_{i}-P_{p}=\frac{4\sigma_{ip}\cos\alpha }{2r}$$
where *σ_ip_* is the equilibrium interfacial tension between the dispersed and continuous phase and *α* is the contact angle. Because the visceral pleura is a hydrophilic membrane, Δ*p* > 0 and *P_i_* > *P_p_* are observed. The two principal mechanisms by which a lung forms a spherical image are shear-controlled detachment induced by shear stress on the membrane surface, and spontaneous detachment induced by a gradient in Young–Laplace pressure [8]. On the contrary, the lung maintains its tapered shape in the case of an active air leak because the continuous air refill does not allow a complete parenchyma re-expansion (Figure 2a,b).

We suggest the hypothesis that extended volume resections (bilobectomy) cause bigger residual pleural spaces, and the combination of these two factors determines the “balloon” sign, which is frequent after bilobectomy, but rarer after standard lobectomy and subanatomical resection.

## 3. Clinical Considerations (Prolonged Air Leak)

The most commonly used method for the qualitative assessment of air leakages is inviting the patient to cough and observe the water column of the drain chamber. If no air bubbles are observed, it means that there is no active air leak in an airtight lung; on the other hand, whenever bubbles are detected in the chamber, it discloses air in the pleural space, without effectively discriminating between active or passive leakage. When the number and intensity of bubbles remain stable at repeated coughs, we may suppose an active air leak; on the other hand, if they decrease—and in some cases stop after few coughs—it is more likely to indicate a small active leakage or a passive one [13].

About half of the operated patients disclose at least minor leakages after pulmonary resection, although the vast majority of these leakages stop two or three days after surgery. On the contrary, persistent air leaks are defined as those lasting from four days to greater than ten days postoperatively [14]. Prolonged postoperative air leaks are considered a complication when they persist for more than five days, and they are the most important determinant of length of postoperative hospital stay [15].

The most common risk factors for prolonged postoperative air leaks are: a low predicted forced expiratory volume in 1 s (FEV 1), upper lobe lobectomies, incomplete or fused fissures, emphysema, extensive pleuro-pulmonary adhesions, infectious diseases, and chronic inflammatory conditions [16,17,18,19,20].

Several options are available for preventing or treating prolonged postoperative air leaks, depending on their entity and aetiology. Intraoperative pneumoperitoneum is a possible option in case of incomplete lung re-expansion after right upper lobectomy [21,22]. The injection of autologous blood into the pleural cavity through the chest drainage is a non-surgical alternative to cause pleurodesis. Although the sclerosing effect of the injected blood might be not as effective as that of chemical sclerosing agents or drugs like sterile talc, it can be indicated in some cases of mild/moderate air leaks because of its property of occluding alveolar leaks by fibrin formation [23,24]. A bronchoscopic approach to prolonged air leaks by endobronchial valve treatment can be indicated in selected patients and might offer effective results [25,26,27]. A Heimlich valve—which is a one-way valve allowing air passing through the valve in one direction (from the chest cavity to external space)—is usually used to reduce the length of hospital stay and to safely discharge patients with a chest tube still in place. It has been reported that the Heimlich valve is an excellent option for the outpatient treatment of pneumothorax [28].

When conservative approaches fail, surgical treatment of prolonged air leaks should be taken into consideration. The minimally invasive video-assisted thoracoscopic approach (VATS) or an open approach allow the application of aerostatic agents under vision, pleurodesis, pleurectomy, and stapling of parenchymal lesions, when indicated [29]. Pleural tenting is another surgical option for preventing postoperative air leaks following upper lobectomies and bilobectomies [30,31]. Buttressing of the staple line is recommended in lung resection in emphysematous patients; on the contrary, in non-emphysematous patients, the use of buttressed staples is not well established [14]. Air leaks represent one of the most common complications after standard pulmonary resection; it prolongs hospital stay, causing patient discomfort and increasing hospitalization costs [32]. Many trials have been carried out to assess the ideal techniques, devices, and methods to prevent or reduce postoperative air leaks, but at least 5% of operated patients still show an active air leak at the time of discharge [6].

In the case of high-risk patients for prolonged postoperative air leaks, additional procedures could be taken intraoperatively to help prevent them. Moreover, the use of a protective ventilation plan by the anaesthesiologists can further support these procedures to minimise the risk of ventilation-dependent barotraumas. In addition, postoperative physiotherapy should be tailored to the patient’s needs, thus preferring volume incentivisation exercises rather than forced expiratory ones and provoked cough.

In our previous experience, the computed tomography (CT) assessment of emphysema was found to be a helpful predictor of prolonged postoperative air leaks, being more accurate than pulmonary function tests [6]. This is probably due to the fact that CT results only rely on the volumetric measurement of emphysema, and are not affected by some dynamic variables conditioning pulmonary function tests results such as pulmonary parenchyma compliance, diaphragm motility, expiratory and inspiratory muscle resistance, and strength.

We usually do not apply postoperative suction to chest drainages to manage air leaks; in fact, on one hand, active suction applied to healthy lung tissue may improve parenchyma re-expansion, creating pulmonary adhesions and limiting air leaks. On the other hand, uninterrupted aspiration—put on emphysematous and fragile lung parenchyma—may extend and worsen air leaks, thus inhibiting rather than supporting effective pleural healing. Given that the majority of operated patients are emphysematous, we do not apply routinely postoperative suction to chest drainages and would rather suggest only applying continuous aspiration in cases of significant active bleeding [6].

We observed that induction chemotherapy did not impact on postoperative prolonged air leaks; on the contrary, it only significantly affected preoperative carbon monoxide lung diffusion capacity (DLCO) because of chemical damage to the alveolo–capillary membrane.

## 4. Clinical Consideration (Residual Pleural Space)

Any residual postoperative intrathoracic space represents a major concern for thoracic surgeons because of possible complications and risks related to this condition [3]. Its incidence was historically reported to range between 22% and 40%, more frequently after lobectomy for infectious diseases and in particular for tuberculosis [33,34]. Although the main cause of residual pleural space was initially thought to be postoperative bronchopleural fistula, later studies ruled out this hypothesis, disclosing a completely benign aetiology consisting of a size mismatch between the residual lung volume and pleural cavity [1,35]. In addition, a long-lasting residual pleural space may significantly impact on the postoperative course and increase morbidity and mortality rates. The main physiological mechanisms that eliminate the dead space after lung resections are: (1) residual lung parenchyma hyperinflation, (2) ipsilateral mediastinal shift, (3) hemidiaphragm elevation, and (4) the reduction of intercostal spaces. On the contrary, the main factors contributing to residual pleural space development are: (1) the volume of the resected pulmonary lobe, (2) the volume of the remaining healthy lung, (3) the compliance of the remaining lung parenchyma, and (4) the underlying pulmonary parenchyma disease.

In a series of studies by Misthos et al., residual pleural space was significantly more frequent in female patients, after right-sided procedures and after prolonged air leaks; on the other hand, benign underlying diseases and lower lobectomies were less frequently complicated by residual pleural space [3]. In order to prevent postoperative residual pleural space, an accurate preoperative assessment of pulmonary compliance is highly recommended, as well as careful intraoperative air leak management, pleural decortication whenever needed, correct chest drainage positioning, and eventually a preventive pleural tent, if indicated [3].

Proper residual pleural space management can lead to the successful elimination of dead space, but, in some cases, a residual, small-sized apical space may persist. It can be filled by trapped pleural effusion, without any clinical impact; on the other hand, fibrothorax may develop in some cases with chest pain and pleural empyema, requiring surgical treatment [3].

Postoperative residual pleural space can be defined as a “dynamic” event: Solak et al., in fact, demonstrated a postoperative incidence of 41.4% on postoperative day 1 and 10.3% 12 weeks after surgery [2]. Moreover, an active medical therapy based on toilette bronchoscopy, thoracic physiotherapy, bronchodilatator therapy, and negative pressure applied to chest drainages contributed to a gradual lung parenchyma reinflation [2].

## 5. Conclusions

Postoperative residual pleural space after lung resection and postoperative air leaks are two different clinical entities requiring completely different approaches. Residual postoperative pleural space is a part of the pleural cavity that is not fully reoccupied by the remaining lung after pulmonary resection. No treatment is needed in the asymptomatic residual pleural space without any persistent air leak, and chest drain removal can be safely planned. On the contrary, an active and prolonged air leak after lung resection is an absolute contraindication to chest drain removal that may culminate in hypertensive pneumothorax, subcutaneous emphysema, and severe respiratory symptoms. In order to further contribute to an appropriate differential diagnosis between these two settings, we propose a radiological sign that is only observed in the case of residual plural space. In this case, in fact, the lung takes the form of a round balloon due to the hyperinflation condition, which is likely governed by the Young–Laplace equation describing the capillary pressure difference sustained across the interface between two static fluids, such as water and air, due to the phenomenon of wall tension.

## Figures and Tables

**Figure 1 cancers-14-03533-f001:**
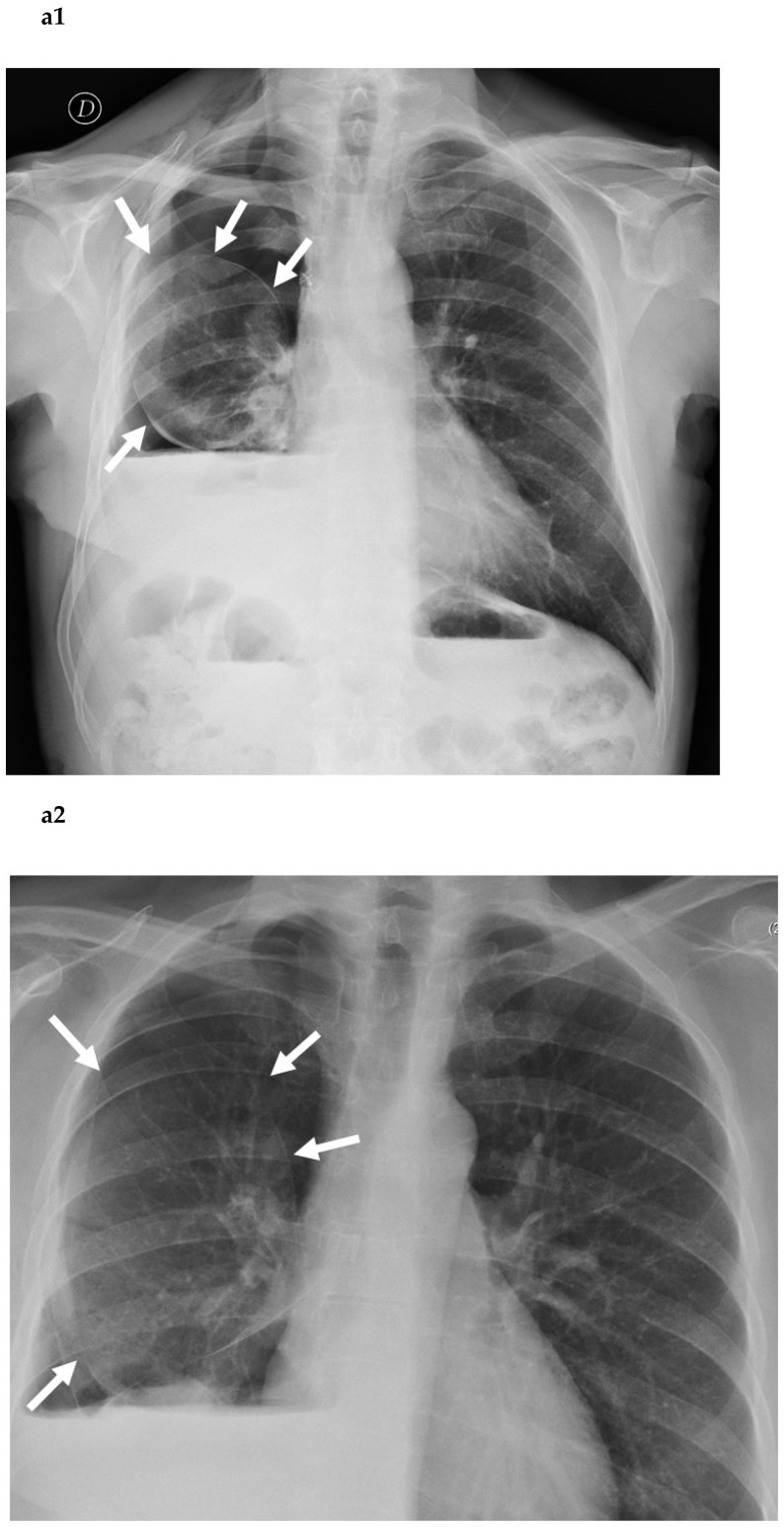
Posterior–anterior chest roentgenogram in the upright position of a patient after lower bilobectomy, showing a rounded well-aerated residual lung parenchyma (white arrows in (**a1**,**a2**)), like an inflated balloon (**b**).

**Figure 2 cancers-14-03533-f002:**
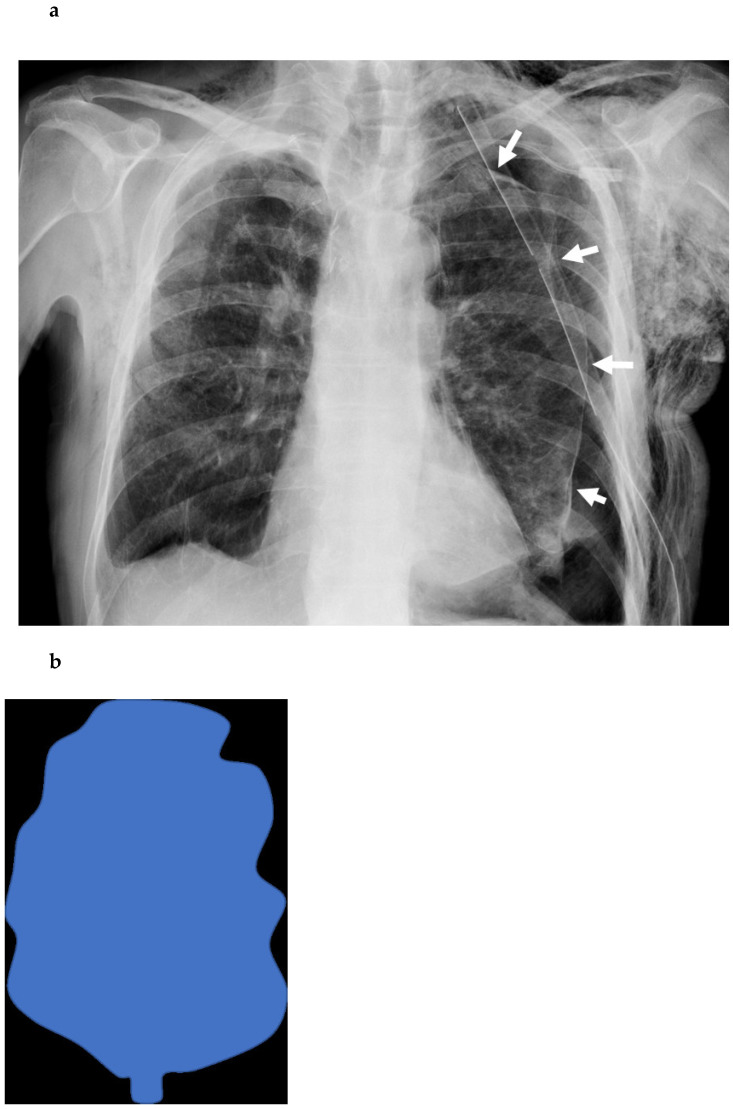
Posterior–anterior chest roentgenogram in the upright position of a patient after left lower lobectomy, showing undulated margins (white arrows) in (**a**), due to the incomplete re-expansion of the lung parenchyma due to air leaks, like a deflated balloon (**b**).

## Data Availability

Available on request.

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
