# Peer review of "The “Balloon-Like” Sign: Differential Diagnosis between Postoperative Air Leak and Residual Pleural Space: Radiological Findings and Clinical Implications of the Young–Laplace Equation"

_cancers, 2022, doi:10.3390/cancers14143533_

Round 1
Reviewer 1 Report
Overall:
- The authors describe a new radiological sign to distinguish between benign post-operative pleural space and active air leak, and discuss the clinical implications of differentiating between these two diagnoses.
- Can the authors comment on how the size of the residual air space is expected to affect the appearance of the remaining lung, and if the type of lung resection (i.e. bilobectomy, lobectomy, wedge resection, etc) is expected to make a difference? It would be useful to see more than just the one example of bilobectomy shown in Figure 1.
Introduction
- The word “indeed” on page 2, line 59 is not necessary.
- I propose changing “In order to further contribute to an appropriate differential diagnosis between post-operative pleural space and active air leak, we propose a radiological sign that is observed only in the first setting.” to “Here, we report a radiological sign that helps to distinguish between post-operative pleural space and active air leak.”
Physical considerations:
- The word “anyway” on page 5, line 128 is not necessary.
Clinical considerations (prolonged air leak):
- There is a fairly extensive discussion on treatment for prolonged air leaks, which is somewhat outside the scope of the paper. I recommend making this discussion shorter, just including enough pertinent information to emphasize that treatment differs between the two conditions explained in the paper.
- I would rewrite the following sentence for clarity: “Intraoperative pneumoperitoneum by injecting about 800 ml of air into the abdomen with a Veress needle can be taken into con-144 sideration mainly in case of incomplete filling of the chest cavity by the remaining lung 145 after upper lobectomies [21,22].” (page 5, lines 143-146).
- Consider changing “sclerotising" (page 5, lines 148 and 149) to “sclerosing”
- Typo on page 6, line 161 “air lekas” should be “air leaks”
- “carried on” (page 6, line 167) should be “carried out”
Clinical considerations (residual pleural space):
- “…later studies ruled out this hypothesis disclosing a completely benign etiology [38,39].” I recommend the authors to explain this “completely benign etiology” rather than just referring to these papers, as it’s relevant in understanding the findings of the paper.
Conclusion:
- I strongly recommend changing “…which is governed…” (page 7, line 242) to “…which is likely governed…”, as you have not proven that this is the explanation for the findings.
Others:
- I answered “Yes” for inappropriate self-citations because, as noted below, there is an unnecessary exhaustive discussion of air leak management, which leads to several self-citations. If this discussion is considered relevant, then the citations may be appropriate.
Author Response
The authors describe a new radiological sign to distinguish between benign post-operative pleural space and active air leak, and discuss the clinical implications of differentiating between these two diagnoses.
- Can the authors comment on how the size of the residual air space is expected to affect the appearance of the remaining lung, and if the type of lung resection (i.e. bilobectomy, lobectomy, wedge resection, etc) is expected to make a difference? It would be useful to see more than just the one example of bilobectomy shown in Figure 1.
We suggest the hypothesis that extended volume resections (bilobectomy) cause bigger residual pleural spaces and the combination of these two factors determines the “balloon” sign which is indeed frequent after bilobectomy but definely rarer after standard lobectomy and subanatomical resections.
Revised version #1, lines 123 - 126
We added a further Chest x ray of another patient disclosing the same radiological sign.
Revised version #1, new figure 1a2 and lines 86,87.
Introduction
- The word “indeed” on page 2, line 59 is not necessary.
We removed the word “indeed”
Revised version #1, line 59
- I propose changing “In order to further contribute to an appropriate differential diagnosis between post-operative pleural space and active air leak, we propose a radiological sign that is observed only in the first setting.” to “Here, we report a radiological sign that helps to distinguish between post-operative pleural space and active air leak.”
We changed as suggested
Revised version #1, line s 67, 68
Physical considerations:
- The word “anyway” on page 5, line 128 is not necessary.
We removed the word “anyway” as suggested
Revised version #1, line 139
Clinical considerations (prolonged air leak):
- There is a fairly extensive discussion on treatment for prolonged air leaks, which is somewhat outside the scope of the paper. I recommend making this discussion shorter, just including enough pertinent information to emphasize that treatment differs between the two conditions explained in the paper.
We shortened the discussion thus removing two self citations (36 and 37)
- I would rewrite the following sentence for clarity: “Intraoperative pneumoperitoneum by injecting about 800 ml of air into the abdomen with a Veress needle can be taken into con-144 sideration mainly in case of incomplete filling of the chest cavity by the remaining lung 145 after upper lobectomies [21,22].” (page 5, lines 143-146).
We rewrote the sentence
“Intraoperative peumoperitoneum is a possible option in case of incomplete lung re-expanison after right upper lobectomy”
Revised version lines 154 - 155
- Consider changing “sclerotising" (page 5, lines 148 and 149) to “sclerosing”
We changed as suggested “sclerotising" to “sclerosing”
Revised version lines 158 - 159
- Typo on page 6, line 161 “air lekas” should be “air leaks”
We changed “air lekas “ to “air leaks”
Revised version line 172
- “carried on” (page 6, line 167) should be “carried out”
We changed “carried on “ to “carried out”
Revised version line 178
Clinical considerations (residual pleural space):
- “…later studies ruled out this hypothesis disclosing a completely benign etiology [38,39].” I recommend the authors to explain this “completely benign etiology” rather than just referring to these papers, as it’s relevant in understanding the findings of the paper.
We added a further phrase to beete explain the underlying mechanism “… consisting in size mismatch between residual lung volume and pleural cavity”
Revised version lines 214, 215
Conclusion:
- I strongly recommend changing “…which is governed…” (page 7, line 242) to “…which is likely governed…”, as you have not proven that this is the explanation for the findings.
We changed as suggested “which is governed…” to “…which is likely governed…”,
Others:
- I answered “Yes” for inappropriate self-citations because, as noted below, there is an unnecessary exhaustive discussion of air leak management, which leads to several self-citations. If this discussion is considered relevant, then the citations may be appropriate.
We shortened the discussion thus removing two self citations (36 and 37)
Reviewer 2 Report
I very enjoyed the authors' fine review on how to make a differential diagnosis between prolonged air leaks and residual space after pulmonary resection. Their work covered all the important and crucial points about this tricky clinical situation very well. As a thoracic surgeon, it appears that they suggested a very useful non-invasive radiological tool to differentiate between prolonged air leaks and residual space phenomenon. I think that the authors' comprehensive and critical review on air leak issue after pulmonary resection should be published in the literature so that the readers can take advantage of this review in clinical practices.
Author Response
REVIEWER 2
I very enjoyed the authors' fine review on how to make a differential diagnosis between prolonged air leaks and residual space after pulmonary resection. Their work covered all the important and crucial points about this tricky clinical situation very well. As a thoracic surgeon, it appears that they suggested a very useful non-invasive radiological tool to differentiate between prolonged air leaks and residual space phenomenon. I think that the authors' comprehensive and critical review on air leak issue after pulmonary resection should be published in the literature so that the readers can take advantage of this review in clinical practices.
Thank you for your evaluation.